# Smith-Magenis Syndrome: Molecular Basis of a Genetic-Driven Melatonin Circadian Secretion Disorder

**DOI:** 10.3390/ijms20143533

**Published:** 2019-07-19

**Authors:** Alice Poisson, Alain Nicolas, Idriss Bousquet, Véronique Raverot, Claude Gronfier, Caroline Demily

**Affiliations:** 1GénoPsy, Reference Center for Diagnosis and Management of Genetic Psychiatric Disorders, Centre Hospitalier le Vinatier and EDR-Psy Q19 Team (Centre National de la Recherche Scientifique & Lyon 1 Claude Bernard University), 69678 Bron, France; 2Laboratoire d’hormonologie-CBPE, CHU de Lyon, 59, boulevard Pinel, 69677 Bron, France; 3Lyon Neuroscience Research Center, Integrative Physiology of the Brain Arousal Systems, Waking Team, Inserm UMRS 1028, CNRS UMR 5292, Université Claude Bernard Lyon 1, Université de Lyon, 69675 Lyon, France

**Keywords:** circadian clock, circadian cycle, melatonin, neurogenetics, *RAI1*, *CLOCK*

## Abstract

Smith-Magenis syndrome (SMS), linked to *Retinoic Acid Induced* (*RAI1*) haploinsufficiency, is a unique model of the inversion of circadian melatonin secretion. In this regard, this model is a formidable approach to better understand circadian melatonin secretion cycle disorders and the role of the *RAI1* gene in this cycle. Sleep-wake cycle disorders in SMS include sleep maintenance disorders with a phase advance and intense sleepiness around noon. These disorders have been linked to a general disturbance of sleep-wake rhythm and coexist with inverted secretion of melatonin. The exact mechanism underlying the inversion of circadian melatonin secretion in SMS has rarely been discussed. We suggest three hypotheses that could account for the inversion of circadian melatonin secretion and discuss them. First, inversion of the circadian melatonin secretion rhythm could be linked to alterations in light signal transduction. Second, this inversion could imply global misalignment of the circadian system. Third, the inversion is not linked to a global circadian clock shift but rather to a specific impairment in the melatonin secretion pathway between the suprachiasmatic nuclei (SCN) and pinealocytes. The development of diurnal SMS animal models that produce melatonin appears to be an indispensable step to further understand the molecular basis of the circadian melatonin secretion rhythm.

## 1. Introduction

The study of the phenotype linked to a single genetic event is an excellent way to improve our knowledge regarding common health issues. In that regard, Smith-Magenis syndrome (SMS) is a formidable approach to better understand circadian melatonin secretion cycle disorders and the role of the *Retinoic Acid Induced* (*RAI1)* gene in this cycle. In 90% of cases, SMS is linked to a microdeletion of chromosome 17 (Figure 1) that encompasses the *RAI1* gene and causes major sleep-wake disorders [1,2,3,4]. Rarely, SMS is caused by a heterozygous mutation in the *RAI1* gene. The estimated frequency of SMS is 1/25,000 but is likely underestimated [5]. The main clinical features of SMS include intellectual disability, dysmorphism and characteristic disturbances of the sleep-wake rhythm. SMS is a unique model of inversion of the circadian melatonin secretion cycle. The study of SMS patients and the recent development of animal models, particularly murine models, should make it possible to further understand the physiopathology of this inversion.

## 2. Description of the Phenotype

### 2.1. Dysmorphism and Visceral Disorders

SMS patients usually present with marked facial dysmorphism. Typically, children have light hair with a bulging forehead, hypertelorism, oblique outer and upper palpebral fissures, synophrys, moderate hypoplasia of the middle floor with the ogival palate, short velum, and velopharyngeal insufficiency. These patients also have short limbs, and scoliosis is frequent, sometimes requiring surgical treatment. In some patients, android obesity occurs during adolescence and may be associated with sleep apnea syndrome. Various congenital heart diseases affect 30% of subjects. Abdominal ultrasonography may reveal splenic malformations and/or renal development and urogenital tract abnormalities [6,7]. Ophthalmological disorders may include iris anomalies, strabismus, and/or microcornea. Refraction abnormalities are also frequent. Hearing impairment is present in about 70 % of subjects of SMS patients with a 17p11.2 deletion [8]. Biological assessment may indicate hypothyroidism, hypercholesterolemia and/or immunoglobulin (Ig) deficiency [6,9].

### 2.2. Neuro-Cognitive Impairment

Intellectual disability with language delay affects almost all children with SMS. Cognitive impairment is variable, usually mild to moderate and rarely severe. Wider deletions result in worse cognitive impairment [10]. Some patients have normal intelligence quotients, and they most often carry the *RAI1* mutation. [11,12] Behavioral disorders are not directly related to the cognitive phenotype. Patients without intellectual disability may sometimes exhibit extremely severe behavioral disorders. These disorders are clearly increased by sleep-wake rhythm disorders [13,14].

The gap between SMS children and other children widens starting at the age of 3. Speech delay, hearing impairment, hyperactivity and attention disorders disrupt academic learning, despite a relative preservation of long-term memory and perceptual abilities [12]. On the other hand, deficits in short-term memory, sequential information processing, and visuomotor, attentional and executive abilities are usually more pronounced [12]. No premature age-related cognitive decline has been observed in this syndrome [15].

Most SMS patients suffer from significant anxiety. Behavioral issues often include outbursts, attention deficit/hyperactivity disorders, self-injury with onychotillomania and polyembolokoilamania (insertion of objects into body orifices), etc. Interestingly, the stronger the speech delay and sleep disorders, the more severe the behavioral issues. Some SMS patients may experience social interaction disorders, including autism spectrum disorder. Interestingly, some authors have proposed that *RAI1* is an autism susceptibility locus [16]. Diagnosis of autism spectrum disorders in SMS patients may be difficult to confirm as the diagnostic scales have not been adapted to syndromic forms of autism. Interestingly, SMS-driven sleep-wake rhythm disorders do not differ regardless of whether the patient presents with or without an autism spectrum disorder.

Morphological, cognitive and behavioral phenotypes may be variable in SMS subjects carrying identical microdeletions or mutations and even between monozygotic twins [17,18]. In contrast, the correlation between SMS and sleep-wake rhythm disorders appears to be robust as sleep-wake rhythm disorders affect virtually all children with SMS regardless of the nature of the genetic event underlying the syndrome.

### 2.3. Sleep-Wake Rhythm Disorders

In early SMS descriptions, the focus was on behavioral disorders and hyperactivity, but there was little mention of sleep disorders [1,2,19]. The first studies on sleep in children with SMS reported that 62% of these children fell asleep, woke up very early and showed sleep maintenance disorders (i.e., frequent nocturnal awakenings) [14]. Rarely, a complete absence of paradoxical sleep is noted [19]. Several studies have confirmed the previously acquired data. These studies also introduced the notion of abnormal chronology of the sleep-wake cycle, which includes falling asleep and waking up early, and the need for several daytime naps [20,21,22]. Therefore, there is a sleep phase advance in SMS. Diurnal alertness usually fluctuates in most patients, and SMS is frequently associated with intense sleepiness around noon.Sleep disorders in SMS have been linked to a general disturbance of the sleep-wake rhythm and coexist with inverted secretion of melatonin [20,23]. The melatonin hormone was isolated in 1958 and is the main hormone secreted by the pineal gland [24]. Melatonin acts as an internal time-giver endocrine messenger [25]. In healthy humans, peak secretion occurs in the middle of the night. In healthy individuals, the melatonin secretion starts at the end of the day, just before sleep onset, peaks in the middle of the night, shortly before core body temperature minimum, and ends at the beginning of the day, slightly after the peak of cortisol secretion.

There are other accessory sources of melatonin in the central nervous system, such as the choroid plexus. Nevertheless, the choroid plexus does not show any circadian pattern [26] and thus does not seem to be involved in the occurrence of the physiological melatonin peak. SMS is characterized by the lack of a melatonin nocturnal peak. In contrast, almost all SMS patients described thus far in the literature show nocturnal residual secretion and a melatonin peak close to noon [20,23]. Therapeutic recommendations consist of the prescription of a beta1-adrenergic antagonist in the morning to avoid the diurnal melatonin peak, along with the prescription of constant release melatonin in the evening [11,12,20,27,28,29,30] to replace the lack of nocturnal autonomous secretion. This therapeutic strategy is likely synergistic, as lower endogenous melatonin secretion results in a greater effect of melatonin medication [31].

The synthesis of melatonin follows an endogenous circadian rhythm driven by the circadian timing system (a.k.a. the circadian clock) located in the suprachiasmatic nuclei (SCN) of the hypothalamus [25]. In addition to internally driven rhythmicity, melatonin is sensitive to light. This light-driven system implicates the retina, retino-hypothalamic tract and SCN. (Figure 2) The SCN relies on the main biological clock of mammals, which generates the body circadian rhythms. To date, several clock genes have been described. These genes control endogenous circadian rhythmicity, and their expression depends on environmental stimuli [32]. Clock gene expression oscillates at a circadian rhythm of approximately 24 h. This rhythm is endogenous with a period close to 24 h, in the absence of synchronizing stimuli [33,34]. Dysfunction of one or more clock genes may be involved in SMS sleep-wake rhythm disorders. SMS is the only known genetic syndrome with an almost constant inversion of melatonin production, and its study represents a great opportunity to better understand the influence of genetic factors on circadian rhythmicity, melatonin secretion and the sleep-wake cycle.

## 3. Genetic Aspects

### 3.1. Deletion 17p11.2

A deletion of approximately 3.5 Mb on chromosome 17p11.2 (Figure 1) underlies 90% of SMS. Typically, this deletion results from chromosomal recombination errors during meiosis (crossing over) according to a non-homologous allelic recombination mechanism [35]. (Figure 3) The critical region includes the *RAI1* gene and extends over 650 kb or less [1,2,3,4,6,36,37]. When included in the deletion, other genes may also contribute to the phenotype. These genes include *PMP22*, which is responsible for some hereditary neuropathies; *TNFRSF13B*, which is involved in immune deficiency; or *MYO15A*, whose deficiency can lead to autosomal recessive non-syndromic deafness [38,39]. These genes are thought to modulate phenotypic features, whereas SMS core symptoms are mainly related to *RAI1* haploinsufficiency [37,40].

In 10% of cases, SMS is due to a heterozygous mutation leading to *RAI1* haploinsufficiency. In these cases, the phenotype is close to the classical SMS phenotype. These patients usually exhibit better cognitive functions. However, sleep-wake rhythm disorders remain identical to those encountered in SMS patients with the classical 17p11.2 deletion [22]. For this reason, the *RAI1* haploinsufficiency is generally considered the core mechanism underlying the inversion of melatonin circadian secretion in SMS subjects. Duplications of the 17p11.2 region have been associated with Potocki Lupski syndrome in which sleep disorders are frequent but not specific [41].

### 3.2. RAI1

Seranski et al. (1999) and Toulouse et al. (2003) mapped *RAI1* to chromosome 17p11.2 using genomic sequence analysis within the critical SMS region [42,43]. This gene is mainly expressed in cerebral tissues, especially in the hypothalamus, cerebral cortex, olfactory bulb, cerebellum and brainstem [40,43,44,45]. Six exons have been described [43]. Most of the coding sequences and pathogenic mutations are located in exon 3 [46], and four protein isoforms have been reported. The promoter region contains binding sites for several regulatory proteins, including a retinoic acid-responsive element [43]. RAI1 proteins contain a polyglutamine domain, two polyserine domains and a zinc-finger domain. The size of the polyglutamine domain does not seem to be involved in the SMS phenotype. In contrast, this expansion may be involved in the age of onset of spinocerebellar ataxia type 2 and might also be involved in the neuroleptic response in schizophrenia [47,48]. Last but not least, zinc-finger domains are also observed in some chromatin binding proteins [49,50]. Thus, RAI1 protein likely carries some chromatin modifying functions and would be able to modulate gene expression [51].

In mice, *Rai1* expression does not show nycthemeral variations [52]. However, this gene, expressed in the hypothalamus, is likely involved in circadian rhythms. Circadian rhythm disorders can have a severe impact on neurodevelopment and more globally, on the health of individuals [53]. Therefore, the dysfunction of one of the circadian clock genes can have a significant impact. In mammals, the SCN is considered the master circadian clock based on lesional studies of these nuclei and the observation that light-induced phase shift and entrainment of behavioral rhythms to the light-dark cycle depend on photic input to SCN neurons [4]. The SCN receives light information and modulates the expression of the main circadian clock genes. In 2012, Williams et al. suggested that RAI1 protein could regulate the circadian expression of the following clock genes: *CLOCK*, *RORC*, *RORA*, *PER3*, *CRY1*, *CRY2*, *NR1D2* and *BMAL*. These authors reported that when the expression of *RAI1* is reduced by 50% in HEK 293T cells, there is a concomitant 50% decrease in *CLOCK* expression. Interestingly, the study of human peripheral blood mononuclear cell transcriptome showed that after four days of simulated night shift work most of the transcripts rhythmically expressed at baseline remained rhythmic with the persistence of the usual phase but with lower amplitudes [54]. Overexpression or *Rai1* in a murine model reveals the reduction of Clock, Per1-2, Arnt1 and Cry1/Cry2 expression in the hypothalamus [41]. Thus, the physiological effect of Rai 1 on clock genes seems gene-dosage dependent [55,56]. A limitation of this kind of study is the difficulty of using data collected from murine, nocturnal models to understand the pathophysiology of sleep in humans. Interestingly, Clock-mutant mice exhibit hyperactivity, decreased sleep, and increased reward-oriented behaviors [57], which, to some extent, is found in SMS patients.

These features have been reported in bipolar patients where elevated melatonin levels have been described during the daytime in manic episodes [58].

As discussed above, *RAI1* is not the only gene involved in the 17p11.2 deletion. Among these genes, *DEXRAS1* might participate in some cases in SMS sleep-wake rhythm disorders. In mice, *Dexras1* appears to be involved in the circadian rhythm but not necessary for its maintenance [59]. Heterozygotes were not evaluated. Further, not all individuals with SMS are heterozygously deleted for *DEXRAS1*. The exact involvement of *DEXRAS1* in human sleep and circadian rhythm remains to be defined.

## 4. SMS and Inversion of the Melatonin Secretion Rhythm

Although the links between *RAI1* and *CLOCK* have become better understood in recent years, the pathophysiology of melatonin profile inversion in SMS remains obscure.

The inversion of the melatonin secretory profile in SMS subjects seems to be a consistent phenomenon. In fact, the inversion was found in 96% of patients in the first SMS cohort (de Leersnyder et al., 2001, eight patients; Potocki et al., 2000, 28 patients) and appears to be pathognomonic for SMS [20,23,60]. In SMS patients, melatonin secretion begins at 6 AM (+/−2 h), and the peak occurs at approximately 12:00 PM (+/−1 h) and lasts approximately 15 h (+/−3.5 h) (compared to controls where melatonin starts at 9 PM +/−2:00, peaks at 3:30 AM +/−1:30, and lasts 10-12 h in controls) (de Leersnyder et al., 2001, eight cases and 15 controls). Studies using actimetry to quantify sleep duration reported a 1 to 2 h decrease in total sleep time/24 h in children with SMS children compared to that in paired controls (de Leersnyder et al., 2001, eight patients and eight controls) [20]. However, cortisol secretion was comparable between the two groups in terms of peak value and time of secretion but was not described otherwise (de Leersnyder et al., 2001, eight patients and 15 controls). The occurrence of moderate to high level of melatonin at noon, has been confirmed in a recent study reporting. The latter suggests that boys present higher melatonin levels at noon than girls [61]. SMS also involves a sleep phase advance [20,62], which is problematic in the daily lives of people with SMS who experience premature falling asleep and early awakenings that are hardly compatible with life in a community.

The link between the diurnal melatonin peak and daytime sleepiness in SMS is well established. In that vein, SMS drowsiness and diurnal behavioral disturbances improve in patients given beta 1 adrenergic-antagonists. This medication blocks the abnormal diurnal melatonin peak via its effect on pineal adrenergic receptors [63]. However, the exact nature of the link between the inversion of the circadian melatonin secretion rhythm and sleep phase advance remains unclear, and no causal link between these two phenomena has been demonstrated thus far. In contrast, the study of rare SMS subjects with a normal melatonin secretion cycle showed that some of them present a sleep phase advance and alteration in the nocturnal sleep structure despite the occurrence of an intact melatonin secretion cycle [64]. The exact nature of the mechanism underlying the inversion of circadian melatonin secretion has hardly been discussed, and only a few studies are available in the literature.

We suggest that three hypotheses could account for an inversion of the circadian melatonin secretion rhythm in SMS patients. To date, none have been systematically tested, and they remain to be investigated.

**Hypothesis** **1:**
*The Inversion of the Circadian Melatonin Secretion Rhythm is Linked to an Alteration in Light Signal Transduction.*


Intriguingly, the circadian melatonin secretion rhythm is regularly inverted and still triggered by dark-light stimulus in SMS. This pattern of circadian melatonin secretion in SMS is different from that observed in most nocturnal animals. Indeed, nocturnal animals usually have a nocturnal peak and no diurnal peak. In these animals, unlike SMS patients, melatonin secretion does not seem to be involved in sleep onset triggering. This pattern is also very different from that observed in most blind individuals or persons deprived of light in which the onset of melatonin and sleep gradually shift according to the duration of the individual period (free run). However, some blind people present with persistent entrainment of the melatonin secretion cycle at 24 h, although with an abnormal phase of melatonin secretion [65] as observed in SMS.

The existence of a light-entrained circadian melatonin secretion rhythm in SMS shows that the sensitivity to light is preserved, although it is altered in a way that appears to produce a reverse effect. RAI1 protein expression is induced by retinoic acid. This vitamin has a major role in the development of the visual system and especially the eyes. Thus, the loss of one *RAI1* allele could induce retinal and light-transduction system alterations. To further explore this hypothesis, Barboni et al. investigated the melanopsin system in five SMS individuals [66]. These authors studied the pupillary response in these subjects and compared their responses with those of four controls. The results revealed an alteration in the sustained component of the pupillary reflex to blue light (470 nm) with a pupillary diameter that normalized faster in patients than in controls after a blue light flash was applied in the dark. In contrast, the response to a red light flash appeared to be preserved in SMS subjects. The authors suggested probable dysfunction of the intrinsically photosensitive retinal ganglion cell (IpRGC)/melanopsin system. A decrease of sensitivity of the melanopsinergic system at the end of the day could explain both an increase in melatonin secretion during the day and an advanced melatonin offset at night, by a combination of a phase advance of the circadian timing system [33] and a decrease photoinhibition during daytime. Given that this system has projections to the SCN of the hypothalamus and inferior olivary pretectal nucleus [66], an impaired sensitivity to light could be involved in the change in phase angle of entrainment that is frequently observed in the circadian rhythm of melatonin secretion in SMS patients. In the same vein, a murine model *Rai 1* −/− has previously revealed a decrease in scotopic electroretinogram responses without any retinal alterations [52].

In addition, data from SMS murine models subjected to circadian protocols are available. Diessler et al. (2017) explored the links between *Rai1*, light exposure and circadian cycle [52]. As mice are nocturnal animals, the results of such studies about light-driven sleep-wake rhythm in mice cannot be directly applied to humans. Furthermore, the C57B11/6 mice used by Diessler et al. do not have detectable melatonin levels over the 24 hours which therefore does not allow investigation into the relationship between *Rai1* haploinsufficiency and melatonin rhythmicity [67,68,69]. However, this work provided several interesting findings. First, *Rai1*+/− mice under dark/light conditions show a drastic decrease in locomotor activity triggered by light exposure. In this model, the physiological residual locomotor activity after light onset is no longer observed in *Rai1* +/− mice, and locomotor activity appears abruptly, as soon as the mice are placed in the dark. Thus, the authors suggested that *Rai+/−* mice present a kind of locomotor hypersensitivity to light. In addition, the authors described a greater phase advance in Rai+/− mice placed under permanent dark conditions than that in wild mice. For mice, light triggers sleep, and interfering with the notion of locomotor hypersensitivity to light in SMS patients should be approached with caution. We suggest that the data of Diessler et al. might be linked to the loss of the transition between sleep and wake, leading to an abrupt onset of sleep and wake states. A loss of residual locomotor activity between awakening and sleep might explain why, in our experience, SMS individuals literally “fall out” of fatigue at the end of the day and are awakened by the daylight at dawn.

No data on the evolution of sleep/wake rhythm and melatonin circadian rhythm in SMS subjects placed on a constant routine are available in humans.

Altogether, these results suggest that an alteration of the IpRGC/melanopsin system and/or of the classical photoreceptors could be partly involved in the abnormal circadian melatonin secretion rhythm and advance of the sleep phase in SMS.

**Hypothesis** **2:**
*The Inversion of the Circadian Secretion Rhythm of Melatonin Implies a Global Misalignment of the Circadian System.*


As SMS involves inverted secretion of melatonin, many animal models have been developed to study the influence of the *RAI1* gene on this cycle [27,41,59,70]. Indeed, individuals with SMS experience some difficulties in maintaining sleep at night, with multiple microawakenings, early awakenings and excessive daytime sleepiness. Excessive daytime sleepiness peaks around noon at the onset of the pathologic melatonin peak.

These clinical features seem consistent with a global circadian disorder. To explore this hypothesis, De Leersnyder (2001) and Potocki (2000) further investigated the circadian parameters of eight (De Leersnyder) and 28 (Potocki) SMS patients [20,23]. Their results confirmed the existence of an inverted circadian secretion rhythm of melatonin. De Leersnyder et al. reported the actimetry data of seven SMS patients. These results confirmed the instability of sleep, the fragmentation of sleep with microarousals, and the presence of frequent naps around noon. De Leersynder et al. also studied the circadian secretion rhythm of cortisol and growth hormone. These authors reported that cortisol followed the usual circadian secretion rhythm and was within the normal range. [20] However, as discussed above, the published data are insufficient to confirm the normality of cortisol secretion, and details about circadian rhythmicity are especially lacking. An intact cortisol secretion rhythm would challenge the hypothesis of a global alteration in the circadian rhythm in SMS. Therefore, SMS would be a singular condition in which the inversion of the circadian secretion rhythm of melatonin would not reflect a global circadian rhythm shift. Furthermore, the internal desynchronization and new phase relation between the rhythms of secretion of cortisol and melatonin could play a role in inducing symptoms and the phase advance of sleep/wake cycle.

**Hypothesis** **3:**
*The Inversion of the Circadian Melatonin Secretion Rhythm Would Not Be Linked to a Global Circadian Clock Shift but Rather to a Specific Impairment in the Melatonin Secretion Pathway between the SCN and Pinealocytes.*


In this hypothesis, the core mechanism underlying the inversion of the secretion rhythm of melatonin would take place downstream of the SCN (Figure 2 and Figure 4). Melatonin synthesis by the pinealocytes of the pineal gland is controlled by a neural system originating from the periventricular hypothalamic nuclei, projecting directly and indirectly to the preganglionic sympathetic neurons of the first thoracic segments of the spinal cord. Then, the postganglionary sympathetic neurons of the superior cervical ganglia project to the pineal gland nerve via the conary nerves [71]. The norepinephrine released by these sympathetic fibers interacts with the beta and alpha noradrenergic receptors located in the membrane of pinealocytes and leads to melatonin production through the activation of the cAMP-PKA-CREB and PLC-Ca++-PKC pathways. Melatonin production is temporary due to its secretion, as there is no storage system in pinealocytes [71,72] (Figure 2 and Figure 4). There is also a retrocontrol pathway modulating the electric and metabolic SCN activities via the activation of specific receptors [73] (Figure 2).

The mechanism underlying the inversion of melatonin circadian secretion would take place somewhere along the pathway between the SCN and pinealocytes. (Figure 4) This mechanism may not be linked to interruption of this pathway, and in that case, we would likely observe no secretion of melatonin at all, which occurs in patients who experience cervical lesions [74]. The diurnal drowsiness would be the visible part of this phenomenon and is improved by the inhibition of pinealocytes’ membranous beta adrenergic receptors [63]. Moreover, if the core mechanism of the inversion of melatonin secretion is downstream of the SCN, light therapy should be ineffective for diurnal sleepiness, which fits well with our clinical experience.

Clock gene dysfunction could be involved in this phenomenon by impairing the initial part of the neural pathway connecting the SCN and pineal gland. Indeed, the genes of the clock system seem to be involved in the light-driven circadian rhythm of melatonin secretion. For example, a haplotype of the *Per 2* gene is associated with a clear decrease in the blocking effect of light on melatonin [75]. The *CLOCK* gene also seems to be involved in the rhythm of melatonin secretion. In this way, an alteration in circadian melatonin secretion rhythm has been observed in zebrafishes placed into dark/dark conditions with selective inactivation of the *Clock* gene in pinealocytes [76], but no inversion of the sleep-wake cycle was noted in these animals. Altogether, these findings are insufficient to suggest that a disturbance of the clock genes governs the inversion of circadian secretion of melatonin in SMS and to rule out an alteration in the transmission of the downstream the SCN. In that vein, Diessler et al. raised the hypothesis of the involvement of the subparaventricular ventral zone (VSPVZ) in the inversion of circadian melatonin secretion rhythm in people with SMS [52]. According to these authors, for nocturnal animals, luminous information is transmitted via direct projections from the retina to the SCN and VSPVZ. The effect of light information on the VSPVZ will induce sleep during the day despite the inhibitory projection from the SCN. In contrast, in diurnal animals, the luminous information is not directly transmitted to the VSPVZ, which receives only inhibitory projections from the SCN. Thus, in diurnal animals, the VSPVZ is inhibited under light conditions, and sleep does not occur. The question of aberrant innervation in SMS, with the occurrence of direct activation of VSPVZ in light conditions, leading to sleep, was raised by these authors. This hypothesis is very interesting; however, it has a major limitation. Indeed, if we imagine a system of light transduction comparable to that in nocturnal animals, how can we explain the occurrence of an inversion of the circadian melatonin secretion rhythm that is not observed in most nocturnal animals?

In brief, in SMS, the mechanism underpinning the inversion of the circadian melatonin secretion rhythm may occur somewhere in the pathway between the SCN and pinealocytes. Further studies are needed to identify this aspect. However, this hypothesis does not seem to explain the advanced phase shift. This phase shift could be linked to a homeostatic disorder, but this hypothesis has not yet been explored per se.

## 5. Could SMS be a Disorder of Homeostasis?

The absence of a nocturnal peak of melatonin disturbs the sleep-wake rhythm. However, this rhythm also depends on homeostatic factors whose influence could become greater to trigger nocturnal sleep in the absence of the physiological peak of melatonin. Sleep homeostasis corresponds to the accumulation of “sleep pressure” during the waking period preceding sleep [77]. Unfortunately, to this day, we are sorely lacking tools to study this aspect of sleep regulation.

The phase advance observed in SMS may be related to a homeostasis disturbance. This disturbance could be either a lowering of the response threshold to homeostatic factors or an abnormal accumulation of homeostatic factors during the day. The abnormal accumulation of homeostatic factors, not counterbalanced by the melatonin cycle, would trigger sleep prematurely during the “evening maintenance arousal phase” and be insufficient to maintain the sleep during the whole night. Clinically, this issue would lead to multiple nocturnal awakenings and early awakening, similar to that observed in the elderly [78].

Tononi and Cirelli suggested that a homeostatic factor may not be a molecule but rather consists of the overall synaptic strength of neural circuits, which increases during wake and renormalizes during sleep [79]. Neuronal plasticity seems to be a major component of sleep regulation homeostasis [80]. Indeed, neuronal plasticity alteration is a recurrent finding in neurodevelopmental disorders and is found in *Rai1* haploinsufficiency [45].

## 6. Conclusions

In SMS, the inversion of the circadian melatonin secretion rhythm and the phase advance of the wake-sleep cycle are two concomitant phenomena that are not necessarily directly related. SMS could constitute a unique genetic model in which the shift in the secretion of melatonin would not reflect a global circadian rhythm disorder. We hypothesize that the inversion of the circadian melatonin secretion rhythm reflects a phenomenon located downstream of the circadian trigger of the SCN along the pathway between the SCN and pinealocytes. The diurnal onset of sleepiness would be the visible and behavioral part of this phenomenon. Additionally, the wake-sleep phase advance might be linked to a disorder of the homeostatic control of the sleep-wake rhythm occurring in SMS. In conclusion, the development of diurnal, melatonin-producing SMS animal models seems to be an indispensable step to further understand the molecular basis of the circadian melatonin secretion rhythm. For example, the Arvicanthis ansorgei which is a diurnal animal model producing melatonin which would be an interesting animal model to further understand the molecular basis of circadian melatonin secretion rhythm. Furthermore, this model would also allow the study of the influence of *Rai1* on this rhythm [81,82,83].

## Figures and Tables

**Figure 1 ijms-20-03533-f001:**
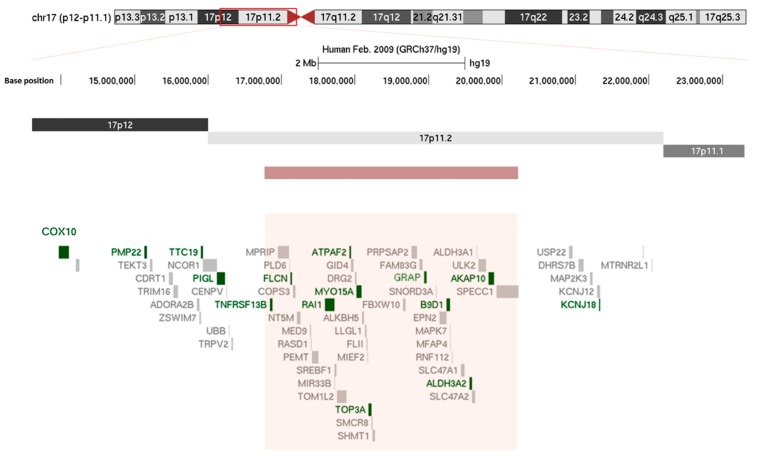
Chromosomic localization of the classical 3.6 Mb 17p11.2 recurrent microdeletion using the UCSC browser (http://genome.ucsc.edu/). The red rectangle localizes to the deleted region. Disease-causing genes are in dark green, while the others are in gray.

**Figure 2 ijms-20-03533-f002:**
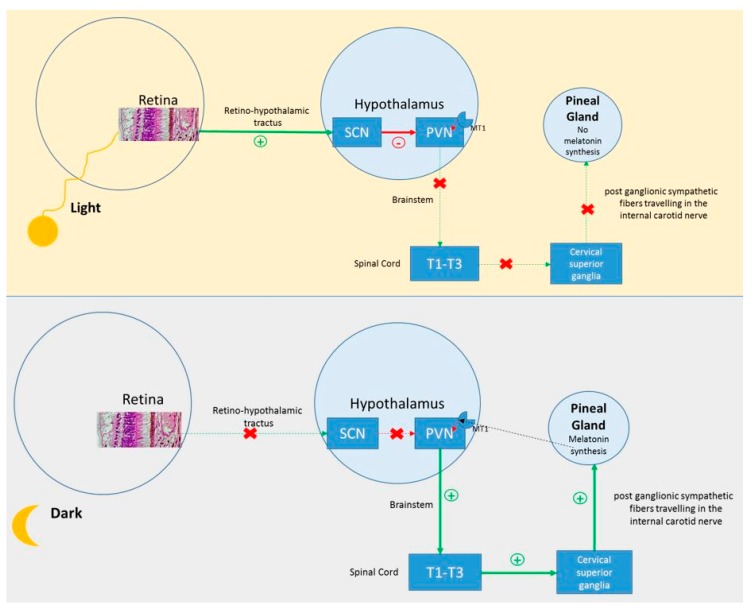
Melatonin synthesis by pinealocytes. The activation of the suprachiasmatic nuclei (SCN) in light conditions inhibits the periventricular nucleus (PVN) and melatonin production. In the absence of light, the PVN is activated and triggers melatonin production through its projections. The dotted line represents the retrocontrol pathway.

**Figure 3 ijms-20-03533-f003:**
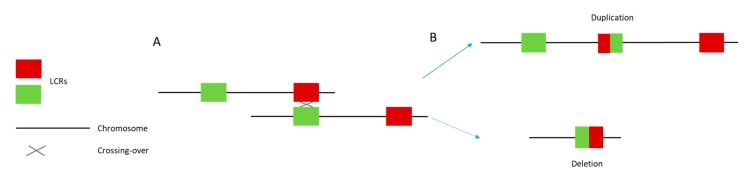
Non-allelic homologous recombination (NAHR) between two chromatids. (**A**) Meiotic misalignment. The gray cross represents unequal crossing over between the two chromatids. This phenomenon is facilitated by the presence of low copy repeats (red and green squares). (**B**) The unequal crossing over leads to the emergence of one gamete with a deleted region and one gamete with a duplicated region.

**Figure 4 ijms-20-03533-f004:**
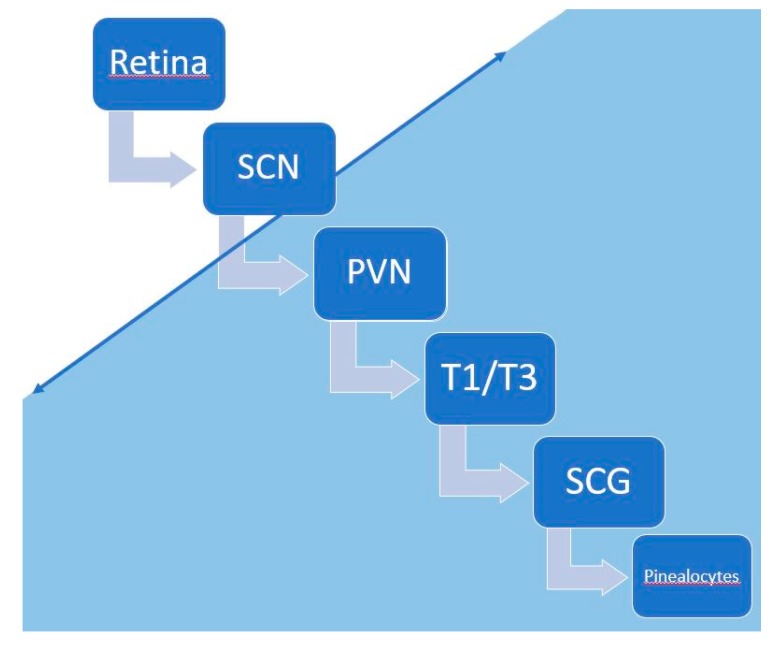
The mechanism underlying the inversion of circadian melatonin secretion occurs in the blue part of the figure, somewhere between the suprachiasmatic nuclei (SCN) and pinealocytes. PVN: periventricular nuclei, T1-T3: preganglionic sympathetic neurons of the first thoracic segments, SCG: superior cervical ganglia.

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
