# Peer review of "Smith-Magenis Syndrome: Molecular Basis of a Genetic-Driven Melatonin Circadian Secretion Disorder"

_ijms, 2019, doi:10.3390/ijms20143533_

Round 1

Reviewer 1 Report

The authors review the sleep and circadian deficits in Smith-Magenis syndrome, a well-known condition with a primary circadian defect that has been studied by a few groups over the years.  The review of the clinical features of SMS is generally complete as background for this disorder, although behavioral problems are well described in this population and they are only briefly mentioned, when they are a primary manifestation of this disorder. 

Several comments that need to be addressed:

1.      Figure 1, GRAP is also a disease gene.

2.     Under neurocognitive development, this statement should have a citation: “Some patients have normal intelligence quotients, and they most often carry the RAI1 mutation.”

3.     Sleep disorder in SMS cannot be described fully as advanced sleep phase --- sleep is not contiguous throughout this period.  Social and cultural ques keep the individual awake as long as they can but they fall asleep often and early, only to wake multiple times in the night.  Finally waking in the morning early---often to fall asleep again in a couple of hours. 

4.     Under Sleep-Wake disorder,Line 36:  circadian rhythm does not forever persist in the absence of stimuli ; free running rhythms are the best example. 

5.     MYO15—autosomal recessive non-syndromic deafness is the correct terminology. Hypoacusia is not adequate.

6.     Figure 3 would benefit from additional text within the figure and legend to explain the duplication and deletion and the clinical outcomes. 

7.     The CAG/polyglutamine repeat in RAI1 is rather constant at 13 or 14 repeats. Very few “expanded” repeats have been described or documented.  The statements related to the CAG repeat need to be corrected.  The occurrence of schizophrenia in association with RAI1 has not been documented; however, one study showed an association with neuroleptic response in this population.  

8.     Under RAI1, RAI1 should be used when referring to human, Rai1 when referring to mouse. (line 26, for example)

9.     The interpretation and discussion of the Williams et al 2012 paper is, overall, incorrect.  

In regards to:  “In 2012, Williams et al. suggested that RAI1 protein could regulate the circadian expression of the following clock genes: CLOCKRORC,RORAPER3CRY1CRY2NR1D2 and BMAL. These authors reported that when the expression ofRAI1 is reduced by 50% in HEK 293T cells, there is a concomitant 50% decrease in CLOCKexpression. However, the authors provided no data on the timeline of the measurements. As the expression of CLOCKPER and BMAL is cyclic, the data provided do not allow us to determine whether the reported decrease in gene expression is real and constant [50]. Furthermore, the circadian period shortening measured by BMAL expression seems modest and has not been reproduced to date. Another limitation of this study is the difficulty of using data collectedfrom murine, nocturnal models to understand the pathophysiology of sleep in humans. In summary, the results presented by William et al. show a link between the decrease in the expression of RAI1 and the expression of CLOCK whose exact nature (decrease, shift…) and consequences remain to be specified.”

According to the methods and data described in this paper, RAI1 and CLOCK were measured at the same time in multiple biological replicates and using multiple sources with consistent results across HEK293T, mouse hypothalamus, and U2OS cells.  In HEK cells, this was done collection of RNA at 24 hours after transfection and it appears the deviation is low, which supports reduced variability at the time of sampling.  Further, mouse hypothalamic were assessed from samples taken in the light and dark phases and results were consistent across tissue, cell cultures, and cells taken from SMS patients and controls-- all assessed together and samples and results are consistent in each group.  This paper also shows that RAI1 binds to a regulatory region of CLOCK and that CLOCK expression is dependent upon RAI1 concentration and that downstream expression of RAI1/Rai1 targets are also affected.  Williams et al showed on multiple levels that RAI1 is a regulator of CLOCK and its connection to circadian abnormalities in SMS.  

The comments about data not being reproducible needs citations as support.  

These papers also discuss similar studies of gene expression related to Rai1 and circadian rhythm anomalies in mouse.  https://www.ncbi.nlm.nih.gov/pubmed/28794907

https://www.ncbi.nlm.nih.gov/pmc/articles/PMC4086898/

These papers also discuss RAI1 related to altered rhythms. 

https://www.ncbi.nlm.nih.gov/pubmed/26384114

10.   The purpose of Figure 4 is difficult to understand.  What is being measured here? Or represented?  This “model” assumes that the entire abnormality is due to a shift in timing of gene expression, when that has never really been shown or proposed in any studies with hard data.  This also assumes a phase shift is all that is happening, but the clinical findings would not support this statement since sleep/wake/rest are not just shifted but disrupted and erratic.  

11.   A potential role for DexRas1 is purely speculative. All studies with that gene in mice have been done in knockourts.  Heterozygotes were not evaluated.  Further, not all individuals with SMS are heterozygously deleted for DEXRAS1 

12.   Biesser et al. (2017) did all studies in Rai1+/- mice.  They did not use knockout mice.  This section needs corrected.

13.   Recommend also discussing/including:  https://onlinelibrary.wiley.com/doi/pdf/10.1111/cns.12653

14.   Lab mice (C57Bl/6, in particular) do not have significant detectable levels of melatonin.  This fact is critical for consideration related to any assessment of melatonin pathways in mice.  

https://www.ncbi.nlm.nih.gov/pmc/articles/PMC2851971/

https://www.physiology.org/doi/full/10.1152/ajpregu.00360.2001

15.  The final statement:  “In conclusion, the development of diurnal, melatonin-producing SMS animal models seems to be an indispensable step to further understand the molecular basis of the circadian melatonin secretion rhythm.”

How do the authors suggest diurnal models will be generated?  They should provide suggestions since this conclusion seems to be the major problem that needs solved.  

16.   The paper as written as multiple fonts within the text; this should be corrected. 

17.   The references are doubly-numbered.  Needs correction.

Reviewer 2 Report

Discussing hypothesis to explain the unique pattern of melatonin rhythm present in Smith Magenis patients is relevant not only for a better understanding of the disease, but also as a model to explore the mechanisms involved in melatonin secretion as well as, in a broader sense, biological rhythms generation and phase relation in humans.  Therefore the present paper fills a gap in gathering some hypotheses related to this issue.

The comments below are intended to add some considerations that might be helpful in expanding the discussion.
